# Taxonomic, Genomic, and Functional Variation in the Gut Microbiomes of Wild Spotted Hyenas Across 2 Decades of Study

Connie A. Rojas,[a,b]* Kay E. Holekamp,[a,b] Mariette Viladomat Jasso,[c] Valeria Souza,[c] Jonathan A. Eisen,[e] Kevin R. Theis[b,d]

aEcology, Evolution, and Behavior Program, Michigan State University, East Lansing, Michigan, USA
bBEACON Center for the Study of Evolution in Action, Michigan State University, East Lansing, Michigan, USA
cInstitute of Ecology, Universidad Nacional Autónoma de México, Mexico City, Mexico
dDepartment of Biochemistry, Microbiology and Immunology, Wayne State University School of Medicine, Detroit, Michigan, USA
eDepartment of Evolution and Ecology, University of California, Davis, Davis, California, USA

**ABSTRACT** The gut microbiome provides vital functions for mammalian hosts, yet research on its variability and function across adult life spans and multiple generations is limited in large mammalian carnivores. Here, we used 16S rRNA gene and metagenomic high-through-put sequencing to profile the bacterial taxonomic composition, genomic diversity, and metabolic function of fecal samples collected from 12 wild spotted hyenas (*Crocuta crocuta*) residing in the Masai Mara National Reserve, Kenya, over a 23-year period spanning three generations. The metagenomic data came from four of these hyenas and spanned two 2-year periods. With these data, we determined the extent to which host factors predicted variation in the gut microbiome and identified the core microbes present in the guts of hyenas. We also investigated novel genomic diversity in the mammalian gut by reporting the first metagenome-assembled genomes (MAGs) for hyenas. We found that gut microbiome taxonomic composition varied temporally, but despite this, a core set of 14 bacterial genera were identified. The strongest predictors of the microbiome were host identity and age, suggesting that hyenas possess individualized microbiomes and that these may change with age during adulthood. The gut microbiome functional profiles of the four adult hyenas were also individual specific and were associated with prey abundance, indicating that the functions of the gut microbiome vary with host diet. We recovered 149 high-quality MAGs from the hyenas' guts; some MAGs were classified as taxa previously reported for other carnivores, but many were novel and lacked species-level matches to genomes in existing reference databases.

**IMPORTANCE** There is a gap in knowledge regarding the genomic diversity and variation of the gut microbiome across a host's life span and across multiple generations of hosts in wild mammals. Using two types of sequencing approaches, we found that although gut microbiomes were individualized and temporally variable among hyenas, they correlated similarly to large-scale changes in the ecological conditions experienced by their hosts. We also recovered 149 high-quality MAGs from the hyena gut, greatly expanding the microbial genome repertoire known for hyenas, carnivores, and wild mammals in general. Some MAGs came from genera abundant in the gastrointestinal tracts of canid species and other carnivores, but over 80% of MAGs were novel and from species not previously represented in genome databases. Collectively, our novel body of work illustrates the importance of surveying the gut microbiome of nonmodel wild hosts, using multiple sequencing methods and computational approaches and at distinct scales of analysis.

**KEYWORDS** animal microbiome, gut, metabolic functions, MAGs, metagenome assembly, social carnivore, hyenas, gut microbiome, longitudinal, mammal microbiome, metagenomics, microbial ecology

Address correspondence to Connie A. Rojas, carojas@ucdavis.edu.

*Present address: Connie A. Rojas, Department of Evolution and Ecology, University of California, Davis, Davis, California, USA.

The authors declare no conflict of interest.

Across mammals, the taxonomic composition of the gut microbiome, specifically the bacterial portion, varies widely among individuals and is often correlated to variations

in diet, reproductive state, age, disease status, habitat, and social interactions of the host (1–10). The gut microbiome can also vary rapidly within hosts over short time scales on the order of days or weeks, which may be due to fluctuations in body temperature, circadian rhythms, a recent meal, or a stressful event (11, 12). Thus, to accurately capture the dynamics of these intestinal communities and their responses to environmental perturbations, longitudinal studies are required. Yet, longitudinal studies of the gut microbiome across host life span are limited, particularly in wild, long-lived mammals. Little is known regarding microbiome trajectories within hosts over time and across hosts within a social group, and also about the relative influences of temporal, ecological, and host factors on these microbiome dynamics. A recent study sampled the gut microbiomes of a population of meerkats (*Suricata suricatta*) from 1997 to 2019 and found that across the meerkat life span, annual variation was the strongest predictor of gut microbiome composition (13). Over shorter time scales, however, host identity outweighed the effects of the other host factors, particularly when samples were collected within 2 months of each other (13). The gut microbiomes of meerkats also showed daily diurnal oscillations, in part due to their temperature-constrained foraging schedules. Another recent study examined the gut microbiomes of baboons (*Papio* spp.) over a 14-year period and concluded that although all gut microbiomes exhibited cyclical seasonal shifts in composition, microbiome dynamics across baboons were only weakly synchronized (14). Instead, baboons exhibited largely individualized gut microbiomes over their life spans, despite their shared diets, environments, and opportunities for between-host microbial dispersal (14). Findings from the two studies suggest that gut microbiome dynamics may be species specific and influenced by host behavior, temporal factors, and habitat characteristics.

Here, we expanded upon this work and conducted a descriptive study to investigate gut microbiome dynamics across host life span and across multiple generations in wild spotted hyenas (*Crocuta crocuta*) over a 23-year period. Hyenas are socially complex, long-lived carnivores (15) whose groups contain multiple overlapping generations of adult females and their young, along with breeding males that have generally immigrated from other clans (15–19). They live in matrilineal societies that are structured by linear dominance hierarchies wherein an individual's social rank, which is not dependent on body size or fighting ability, determines access to resources and fitness (15, 20–23). Hyena societies are also characterized by strong fission-fusion dynamics. Individuals mainly travel, rest, and forage alone or in small subgroups ("fission") that can "fuse" into larger groups (20, 24, 25). The spotted hyenas from our study population reside in the Masai Mara National Reserve (MMNR), Kenya, a savanna habitat that supports high densities of herbivores and carnivores (26). This habitat had undergone significant changes to its landscape and wildlife over the 23-year study period. A consistent decline over time in the densities of wild herbivores and lions was typically accompanied by an exponential increase in anthropogenic disturbance and livestock grazing within the reserve (19, 27, 28). These changes have also coincided with increases in hyena population sizes (29). Given the complex social and ecological dynamics of wild hyenas, high variability might be expected within and between their gut microbiomes.

As has been documented for other mammals (30, 31), we might expect the functions of the gut microbiome of hyenas to be important. For example, in Malayan pangolins (*Manis javanica*) and potentially other myrmecophagous (i.e., ant- and termite-eating) mammals, the microbiome gene repertoire includes over 114 gene modules related to the synthesis of chitin-degrading enzymes and the digestion of insect exoskeletons (32). Stephen's woodrats (*Neotoma stephensi*), which are dietary specialists of juniper, harbor genes associated with the detoxification of juniper secondary metabolites in their gut microbiomes (33). Although here we could not specifically test whether gut microbiome functions directly contributed to hyena physiology, we have provided a global overview of the gene pathways present in the microbiome of four adult females and determined whether the overall gene repertoire varied with hyena prey densities over short timescales. Although hyenas are generalist carnivores and can eat all parts of their prey, including the skin, muscle, bone, and viscera, their diet

**TABLE 1** Longitudinal fecal samples collected from 12 adult wild spotted hyenas, 1993 to 2016[a]

| Matriline and rank | Hyena ID | No. of samples | Years represented by samples | Age (yrs) during study period |
|---|---|---|---|---|
| M1, high rank | M1 | 13 | 1993–1999 | 10–16 |
| | D1 | 35 | 1999–2007, 2011 | 2–14 |
| | G1 | 33 | 2003–2010, 2012–2015 | 2–14 |
| M2, medium-high rank | M2 | 33 | 1997–2007, 2009–2012 | 3–17 |
| | D2 | 24 | 2006–2007, 2009–2014, 2016 | 2–13 |
| | G2 | 17 | 2011–2016 | 2–7 |
| M3, medium-low rank | M3 | 14 | 1993–1995 | 8–10 |
| | D3 | 49 | 1995–2009, 2011–2012, 2015 | 2–22 |
| | G3 | 16 | 2011, 2013–2016 | 3–8 |
| M4, low rank | M4 | 18 | 1993–1995, 1997–2000 | 6–12 |
| | D4 | 27 | 1994–1997, 2000–2004, 2006 | 2–14 |
| | G4 | 24 | 2006, 2008–2014 | 3–11 |

[a]Fecal samples were collected in this study for each of 12 individual adult female hyenas. Also shown is other information about each individual (age, hyena ID, matriline, and rank) as well as the years the samples were collected. For each of four matrilines (M1 to M4), there were samples from three generations (M, mother; D, daughter; G, granddaughter). See Table S1 for a breakdown of samples by year for each individual hyena.

may fluctuate throughout the year. For example, during July through October of every year, hyenas feast on large migratory herds of wildebeest (*Connochaetes taurinus*) and zebra (*Equus quagga*) (34, 35) arriving from the Serengeti. However, during the months when ungulate numbers are low, hyenas scavenge more and consume lower-quality food, such as skin, bones, or old dried animal carcasses that were virtually ignored when fresh food was plentiful (34). The fluctuating prey densities influence the quantity, quality, and type of food hyenas will eat, which has fitness consequences in this species (15, 20–23). Thus, we examined whether the gut microbiome gene repertoire in wild hyenas was associated with ungulate prey densities.

To summarize, we used 16S rRNA gene sequencing of fecal samples to profile gut microbiome composition in 12 adult female hyenas over a 23-year study period, and we used metagenomic sequencing to profile gut microbiome function from 4 adult females over two distinct 2-year periods. The study's sampling collectively spanned three generations of female hyenas: mothers, daughters, and granddaughters (Table 1; see also Table S1 in the supplemental material). All sampled individuals were members of a single social group but belonged to distinct maternal lineages (matrilines) that varied in their social rank. Individuals ranged in age from 2.4 to 22 years over the study period, spanning these animal's natural adult life spans (Table S2). Employing opportunistic sampling, we collected on average 25 samples from each of these females across 9.8 years (range, 13 to 48 samples per hyena) with a median of 3.4 years between consecutive samples (Table 1; Table S1). More frequent sampling of feces from wild hyenas was constrained by the fission-fusion dynamics and large home ranges of this species, which may mean long periods of time between sightings of the same individual. Based on this data set, we assessed the taxonomic variation present in the gut microbiomes of hyenas over the 2 decades of sampling and examined whether any of the variation could be explained by host factors, including individual identity, matriline, age, prey densities, and calendar year. We also identified the gut microbial taxa that constituted the "core" gut microbiome in wild spotted hyenas and consistently persisted over host life spans. These taxa may be functionally important to gut health and animal function (36, 37). Third, we examined the microbiome gene repertoire of four adult females over 2 years and investigated whether it varied with host prey densities, host identity, year, or matriline. Finally, we report the first metagenome-assembled genomes (MAGs) recovered from hyenas, expanding on what is known about the taxonomic and genomic diversity of the mammalian gut, particularly in less-studied species like hyenas. Collectively, our findings provide a novel perspective on the variability, genomic diversity, and function of the gut microbiome in a wild African carnivore by using multiple sequencing types over long and short temporal scales.

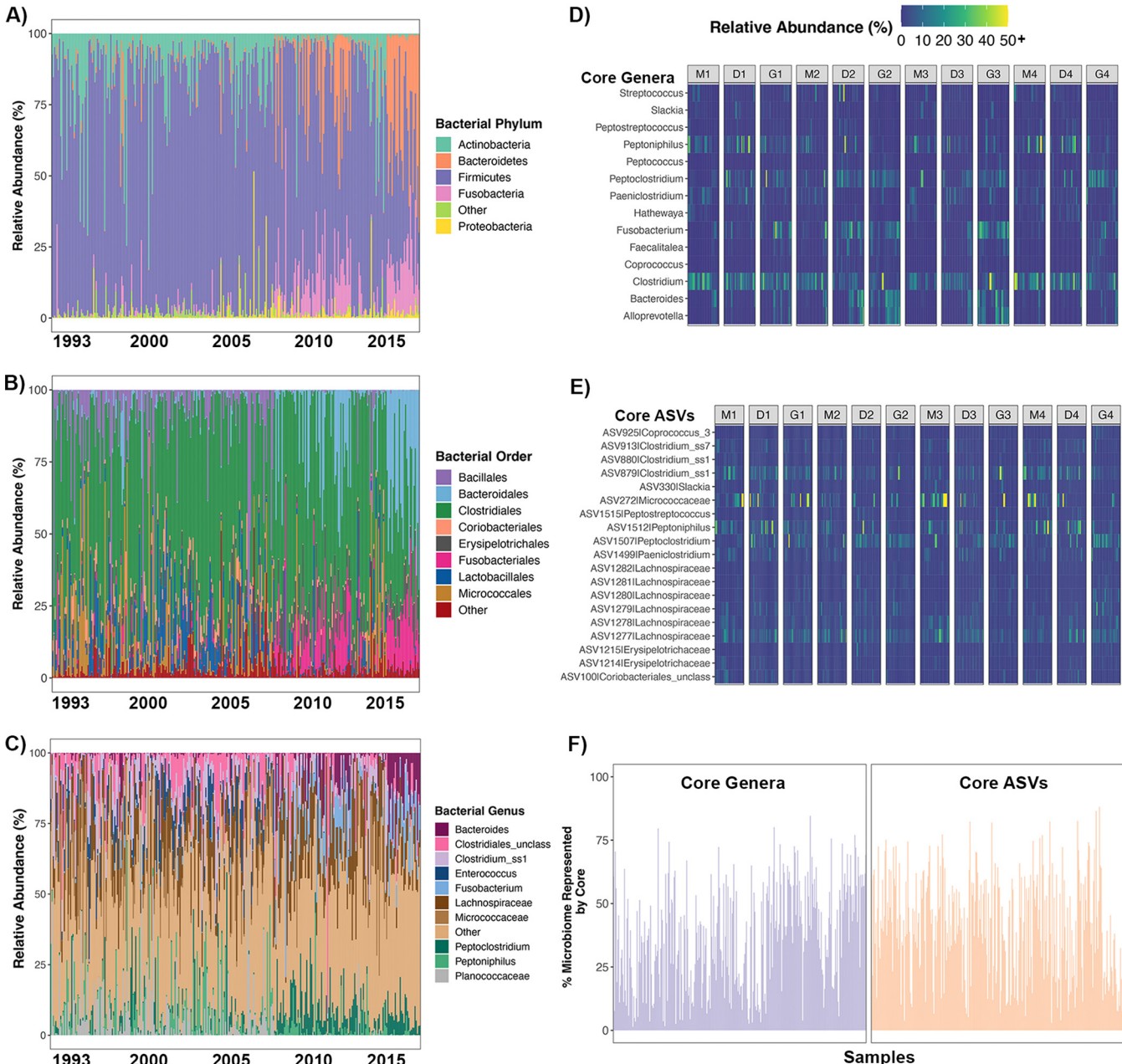

**FIG 1** Amid global and temporal shifts in gut microbiome composition, a taxonomic core was present in the guts of all studied hyenas. (A to C) Stacked bar plots show the relative frequencies of 16S rRNA gene sequences assigned to each bacterial phylum (A), order (B), and genus (C) across samples. Samples are ordered by sampling date, and each color represents a bacterial phylum, order, or genus. (D and E) Heatmap of the relative abundances of 14 core bacterial genera (D) or 19 core bacterial ASVs (E) across samples. These bacterial taxa were found in 85% of samples. Not all sequences were classified to genus or species level, and in those scenarios, the last known classification (e.g., family) was used. (F) Proportion of the microbiome represented by core genera (purple) or ASVs (orange) in each sample.

## RESULTS

**Global shifts in composition of the gut microbiome in wild hyenas sampled across 23 years.** We first investigated variation in 16S rRNA gene profiles over the 23-year study period and evaluated whether any of the variation present could be explained by host endogenous and/or exogenous factors. Across the 12 sampled individuals (see Table S2 in the supplemental material), we found that gut microbiomes showed high interindividual, interannual, and global shifts in composition. The relative abundances of all five dominant bacterial phyla (*Firmicutes*, *Actinobacteria*, *Bacteroidetes*, *Proteobacteria* and *Fusobacteria*) varied over the 2 decades of sampling (Fig. 1A). Additionally, in 2008 to 2009, there was

**TABLE 2** Core bacterial genera and ASVs that were present in >85% of fecal samples from spotted hyenas[a]

| Taxonomic level | Core taxa |
| --- | --- |
| Genus | *Alloprevotella* |
| Genus | *Bacteroides* |
| Genus | *Clostridium* |
| Genus | *Coprococcus* |
| Genus | *Faecalitalea* |
| Genus | *Fusobacterium* |
| Genus | *Hathewaya* |
| Genus | *Paeniclostridium* |
| Genus | *Peptoclostridium* |
| Genus | *Peptococcus* |
| Genus | *Peptoniphilus* |
| Genus | *Peptostreptococcus* |
| Genus | *Slackia* |
| Genus | *Streptococcus* |
| ASV | ASV272, *Micrococcaceae* |
| ASV | ASV127, *Lachnospiraceae* |
| ASV | ASV128, *Lachnospiraceae* |
| ASV | ASV1278, *Lachnospiraceae* |
| ASV | ASV1279, *Lachnospiraceae* |
| ASV | ASV100, *Coriobacteriales*_unclassified |
| ASV | ASV330, *Slackia* |
| ASV | ASV1499, *Paeniclostridium* |
| ASV | ASV1507, *Peptoclostridium* |
| ASV | ASV1512, *Peptoniphilus* |
| ASV | ASV1515, *Peptostreptococcus* |
| ASV | ASV1215, *Erysipelotrichaceae* |
| ASV | ASV913, *Clostridium* |
| ASV | ASV1214, *Erysipelotrichaceae* |
| ASV | ASV879, *Clostridium* |
| ASV | ASV880, *Clostridium* |
| ASV | ASV1280, *Lachnospiraceae* |
| ASV | ASV925, *Coprococcus* |
| ASV | ASV1281, *Lachnospiraceae* |

[a]Bacterial genera or ASVs were considered part of the 'core' microbiome if they were found in at least 85% of all fecal samples used in this study. 16S rRNA gene sequences were classified and assigned taxonomy with the SILVA rRNA gene reference database (v132). Not all sequences were classified to the genus level or species level, and in those instances, their last known classification (e.g., family) was used. In addition to their taxonomic label, ASVs also have a random numerical ID label. For data on their abundances, see Fig. 1D and E.

a marked shift in the gut microbiome compositions of all sampled hyenas. Gut microbiomes which were previously dominated by *Firmicutes* (mainly *Clostridiales*) and *Actinobacteria* (mainly *Bacillales*) came to be dominated by *Bacteroidetes* (*Bacteroidales*) and *Fusobacteria* (*Fusobacteriales*) and contained fewer *Firmicutes* (Fig. 1A and B). *Bacteroidales* and *Fusobacteriales*, which had a combined mean relative abundance of <4% before 2008, became much higher in relative abundance after 2008 and constituted on average 17.27% and 11.24% of the microbiome, respectively (Fig. 1B). In turn, the *Bacillales* mean relative abundance decreased from 10.66% prior to 2008 to 1.37% over the remaining years. At the genus level, the shift appeared to be driven by increases in the relative abundances of *Fusobacterium*, *Bacteroides*, and *Peptoclostridium* and by decreases in the relative abundances of *Enterococcus*, unclassified *Planococcaceae*, and unclassified *Clostridiales* during the years following 2008 (Fig. 1C). Thus, it is evident that the composition of the gut microbiome in 12 wild adult hyenas varied greatly across the 2 decades of sampling, and marked shifts occurred in 2008 to 2009 that were apparent in all hyenas sampled at that time.

Despite the high degree of temporal variability observed in 16S rRNA gene profiles, there were 14 bacterial genera out of 326 genera that represented the core and were present in at least 85% of gut microbiome samples (Table 2; Fig. 1D). Core bacterial genera included *Alloprevotella*, *Bacteroides*, *Clostridium*, *Fusobacterium*, *Paeniclostridium*, *Peptoclostridium*,

*Peptoniphilus*, and *Streptococcus*, among others (Table 2; Fig. 1D). All but five of those genera were found at mean relative abundances of >1% across samples. These 14 core genera collectively constituted ~36% (±20.2% standard error) of the gut bacterial community in any given sample (Table 2; Fig. 1F), although there were several samples throughout the study period that lacked most of the core taxa. Furthermore, 19 out of 1,689 total bacterial amplicon sequence variants (ASVs) were present in over 85% of gut microbiome samples (Table 2; Fig. 1E). Eight of the 19 core ASVs were assigned a genus, and for the remaining ASVs, no information beyond family was known (Table 2; Fig. 1E). Ten of the 19 core ASVs were found at mean relative abundances of >1% across samples. These 19 core ASVs collectively constituted ~40% (±20.1%) of the gut microbiome (Fig. 1F). Again, there were several samples from certain years that did not contain many of the core ASVs. Interestingly, although the collective relative abundance of core microbial taxa was very low in some samples (2 to 4%), it was never zero. Overall, these results indicate that a core microbiome is present in hyenas, but as was observed in the broader 16S rRNA gene profiles, the relative abundance of this core microbiome is labile.

**Host socioecological factors predict distinct aspects of the gut microbiome.** We also examined whether the alpha- and beta-diversities based on 16S rRNA gene profiles were associated with host endogenous and exogenous factors, including individual identity, matriline, age, mean monthly prey abundance, and sample year. Individual identity most strongly correlated with gut microbiome richness, evenness, and phylogenetic diversity (generalized linear model [GLM] likelihood ration test [LRT] $P < 0.05$) (Fig. 2A; Data Set S1, Sheet 1). Additionally, mean monthly prey abundances were negatively correlated with gut microbiome richness; gut microbiomes were marginally less diverse during months of high prey availability than during periods of prey scarcity (GLM LRT $P = 0.019$) (Fig. 2C; Data Set S1, Sheet 1). Host maternal lineage and calendar year were not significantly associated with gut microbiome alpha-diversity (GLM LRT $P > 0.05$). We also reevaluated the effects of the host predictors after accounting for the repeated sampling of individuals (i.e., hyena identity) and temporal variation (i.e., sample year). In these models, gut microbiome richness and phylogenetic diversity varied with host age; gut microbiome diversity tended to be lower in older than in younger adult hyenas (GLM LRT $P < 0.05$) (Fig. 2B; Data Set S1, Sheet 2). Furthermore, as was observed in the earlier models, mean monthly prey abundance was negatively correlated with gut microbiome richness (Data Set S1, Sheet 2).

When we examined gut microbiome beta-diversity, results showed that hyena identity accounted for up to 11.3% of the variance (weighted Unifrac permutational multivariate analyses of variance [PERMANOVA]) (Table 3; Fig. 2D), suggesting that gut microbiomes are individualized to some extent and may be consistent over an adult's life span. Hyena age and matriline accounted for an additional 4.3% and 2.5% of the variation, respectively (Table 3; Fig. 2D). Mean monthly prey abundance and sample year explained little to none of the variation in microbiome beta-diversity (Table 3; Fig. 2D). In principal-coordinates analysis (PCoA) ordination plots, gut microbiome profiles clustered moderately by sample year and did not appear to cluster by the remaining predictors that were evaluated (Fig. S1).

We also ran linear mixed models to determine whether any of the host factors correlated with the relative abundances of core bacterial genera or ASVs. The models accounted for variation attributable to host individual identity and tested core taxa that had mean relative abundances of at least 1%. The relative abundances of five core genera varied with sample year and increased over time [linear mixed model (LMM) $P$ (adj), $<0.05$] (Data Set S1, Sheet 3). These genera were *Alloprevotella*, *Bacteroides*, *Peptoclostridium*, *Fusobacterium*, and *Faecalitalea* (Fig. 2E). *Clostridium* relative abundances were marginally positively correlated with mean monthly prey availability (Fig. 2E). These genera appeared to have higher relative abundances in the hyena gut during periods of high prey availability than during periods of prey scarcity. None of the bacterial genera varied with host age, and only *Clostridium* abundances varied among maternal lineages (Data Set S1, Sheet 3). Hyenas belonging to matriline 3 harbored lesser abundances of this genera compared to hyenas from the highest-ranking

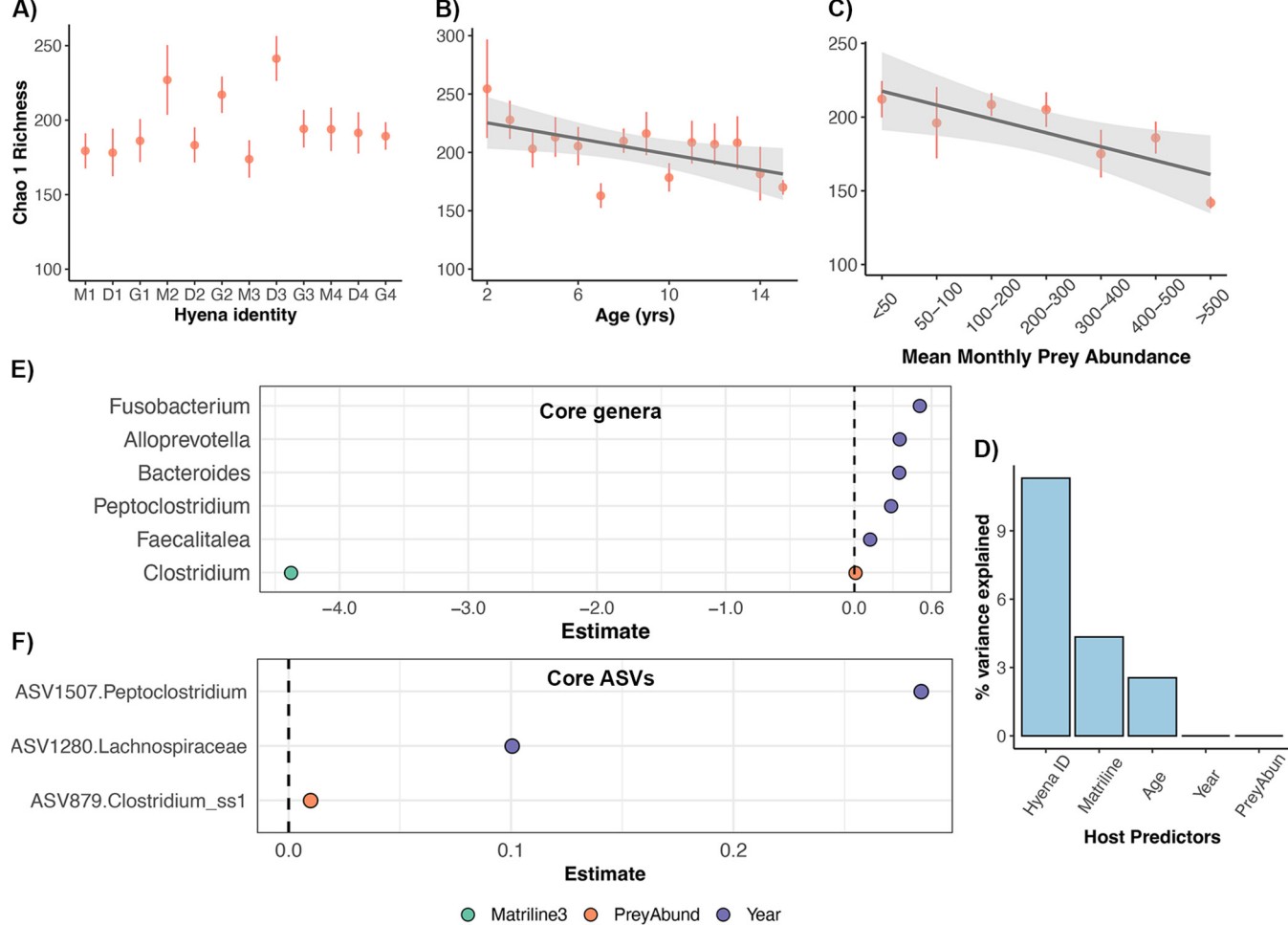

**FIG 2** Socioecological predictors of the gut microbiome in wild spotted hyenas. (A to C) Plots of microbiome Chao 1 richness (mean ± SE) for each hyena individual (A), age category (B), or mean monthly prey abundance (C) for 16S rRNA gene data. The shaded lines with a 95% confidence interval (CI) represent the relationship between $x$ and $y$, estimated as a linear regression. See Data Set S1 in the supplemental material for statistical output from the linear models. (D) Plots of $R^2$ values from PERMANOVA models testing whether host individual identity, matriline, age, mean monthly prey abundance, and sample year predicted gut microbiome beta-diversity (for 16S rRNA gene data). See Table 2 for exact model output. (E and F) Plots of beta-coefficients from linear mixed models regressing the abundance of core genera (E) or ASVs (F) against the five host predictors listed above. Each point represents a beta-coefficient and is color coded by host predictor. If a beta-coefficient is positive, the abundances of that bacterial taxon were positively correlated with the host predictor. Only beta-coefficients with $P$ values of <0.05 after correcting for multiple comparisons are displayed.

matriline. Finally, when conducting a similar analysis on the relative abundances of core ASVs, we found that two ASVs were temporally variable, one ASV was weakly predicted by prey abundance, and none was associated with host age or maternal lineage (Fig. 2F; Data Set S1, Sheet S3).

Collectively, our results indicated that individual signatures in gut microbiome alpha- and beta-diversities were observed for the 12 sampled hyenas. Gut microbiome profiles also varied temporally, suggesting that they are variable within hosts across life span and across generations. Prey abundance was not significantly associated with much variation in 16S rRNA gene profiles, yet differences may exist in microbiome functional profiles, which we discuss below.

**Gut metagenome taxonomic profiles across individuals.** We characterized the gut metagenomes of four individual hyenas, comprising two mother-daughter hyena pairs, using shotgun metagenomic sequencing (Table S2). The pairs belonged to matrilines 1 and 3, and samples from each individual hyena spanned 2 years ($n = 8$ samples per individual, for a total of 32 samples). Putative sequences were classified and assigned taxonomy using Kraken2. Metagenomic sequences were overwhelmingly classified as bacteria (88% mean relative abundance), and a smaller fraction was assigned to viruses and archaea (combined mean relative abundance of 0.56%) (Fig. 3C).

**TABLE 3** Predictors of gut microbiome beta-diversity in adult female hyenas[a]

| Predictor | % Variance, explained by: | | | Avg % variance |
|---|---|---|---|---|
| | Jaccard | Bray-Curtis | Weighted Unifrac | |
| Hyena identity (categorical) | 6.13 ($P = 0.001$) | 8.75 ($P = 0.001$) | 11.31 ($P = 0.001$) | 8.73 |
| Matriline (categorical) | 2.01 ($P = 0.001$) | 2.23 ($P = 0.001$) | 2.55 ($P = 0.001$) | 2.26 |
| Age (yrs) | 1.64 ($P = 0.001$) | 2.41 ($P = 0.001$) | 4.34 ($P = 0.002$) | 2.8 |
| Prey monthly | 0.57 ($P = 0.006$) | 0.44 ($P = 0.10$) | 0.27 ($P = 0.40$) | <1% |
| Yr | 0.64 ($P = 0.001$) | 0.52 ($P = 0.026$) | 0.81 ($P = 0.11$) | <1% |
| Residuals | 88 | 85 | 81 | |

[a]Shown are the $R^2$ values (percent variance explained) and $P$ values for marginal PERMANOVA tests that evaluated whether hyena endogenous and exogenous factors predicted gut microbiome similarity (based on ASV abundances calculated from 16S rRNA gene data). The models utilized 3 types of distance matrices and included five host factors (hyena identity, matriline, age, mean monthly prey abundance, and year). $n = 301$ samples.

We examined the bacterial portion of the Kraken2 results in greater detail. When focusing solely on bacteria, *Firmicutes* (57.9% mean relative abundance across samples) was the dominant phylum, followed by *Actinobacteria* (19.4%), *Fusobacteria* (8.4%), *Bacteroidetes* (8.01%), and *Proteobacteria* (5.09%). All other phyla appeared at mean relative abundances of <1%. The bacterial orders with the highest relative abundances were *Clostridiales* (34.41% mean relative abundance), *Lactobacillales* (13.92%), *Micrococcales* (11.68%), *Fusobacteriales* (8.41%), *Bacillales* (8%), and *Bacteroidales* (6.77%), although the relative abundances of these bacterial orders varied among host matrilines (Fig. 3A). Hyenas from the lower-ranking matriline (M3) appeared to contain greater relative abundances of *Bacteroidales*, *Lactobacillales*, and *Enterobacteriales* than hyenas from the higher-ranking matriline (M1), whose gut microbiomes instead contained more *Clostridiales* and *Fusobacteriales* (Fig. 3A). At the genus level, gut metagenomes mostly contained *Clostridium* (17.88% mean relative abundance), *Enterococcus* (10.64%), *Fusobacterium* (8.29%), *Arthrobacter* (5.28%), and *Bacteroides* (3.38%) (Fig. 3B).

**Housekeeping bacterial functions were the most represented in the gut microbiomes of wild spotted hyenas.** The third aim of this study was to examine the gene repertoires present in the gut microbiomes of four adult hyenas ($n = 32$ samples) and determine whether they were correlated with host socioecology (individual identity, matriline, or prey density) or calendar year. The functional databases used to annotate predicted genes in the metagenomic data were the Clusters of Orthologous Groups (COGs) and the Kyoto Encyclopedia of Genes and Genomes (KEGG).

Not surprisingly, the COG categories with the highest relative abundances were housekeeping functions that are essential for bacterial growth and replication. The categories were ribosomal structure and biogenesis, amino acid transport and metabolism, carbohydrate transport and metabolism, transcription, cell membrane biogenesis, and DNA recombination and repair (Fig. 3D; Data Set S2, Sheet 1). COG pathways, which are less broad than COG categories, were those involved in the synthesis of the bacterial ribosome (50S or 30S subunits), aminoacyl tRNA synthetases (which attach amino acids to tRNA), fatty acids, purines, the amino acid lysine (for protein and cell wall synthesis), and peptidoglycan (cell wall component) (Data Set S2, Sheet 2). Other COG pathways identified were those involved in tRNA modification, glycolysis, and pyrimidine salvage. Similarly, in KEGG profiles, housekeeping functions were the most abundant across samples (Data Set S2, Sheet 3).

We then tested whether the relative abundances of COG categories, COG pathways, or KEGG proteins were associated with host identity, matriline, prey densities, or sample year. Overall, gut microbiome functions were individual specific, as host identity explained ~13% of the variation (Fig. 3E; Table S3 [KEGG protein relative abundances only]). Gut microbiome functional profiles were moderately correlated with prey abundance, and this factor explained ~6% of the variation (Fig. 3F; Table S3 [KEGG protein relative abundances only]). Host matriline and sample year did not explain any variation in metabolic functions. Nearly identical results (unpublished data) were obtained when correlating COG or KEGG abundances with the abundances of metagenome

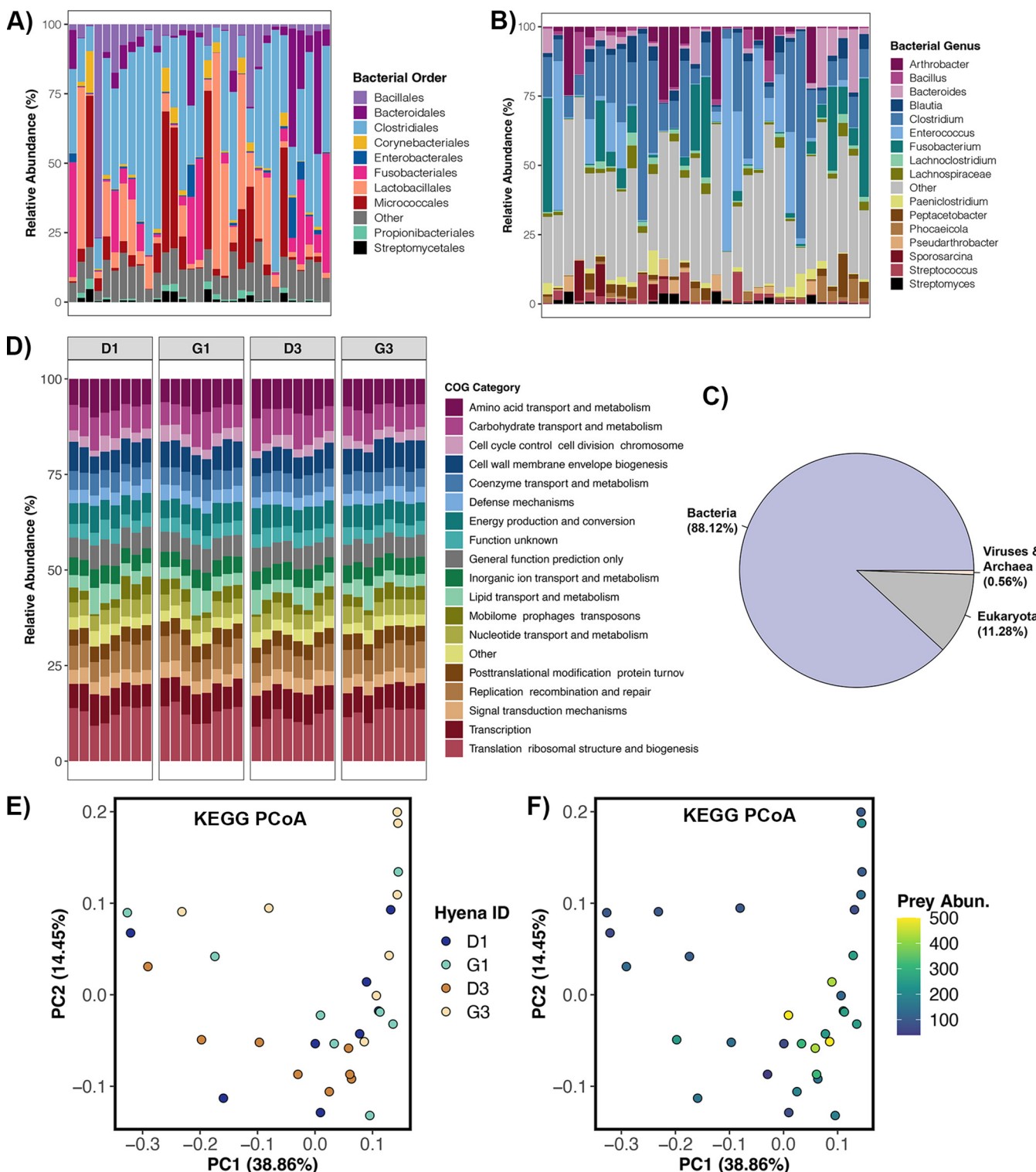

**FIG 3** Composition and predicted function of gut metagenomes obtained from shotgun metagenomic data. (A and B) Stacked bar plots show the relative frequencies of shotgun metagenome sequences assigned to each bacterial order (A) and genus (B) as determined by Kraken2. Samples are organized by hyena individual. (C) Proportion of metagenome sequences assigned to bacteria, eukaryota, and archaea and viruses as calculated by Kraken2. (D) Relative abundances of the most represented COG functional categories across samples. (E and F) PCoA ordinations based on KEGG relative abundances, color-coded by host individual identity (E) or mean monthly prey abundance (F). Contigs were assembled from metagenomic data and imported into Anvi'o for gene prediction and functional annotation. Salmon was used to calculate the relative abundances of genes in each sample (in TPM), and these values were converted to proportions (i.e., relative abundances).

sequences that matched to the genomes of 10 common hyena prey species, the latter of which were calculated with the sourmash software package (38).

To summarize, our findings suggested that gut microbiome functional repertoires in the four sampled hyenas may be influenced by host prey densities and exhibit individual-specific signatures. No functional pathways or enzymes were directly related to the digestion of bone (e.g., phosphatases, collagenases), although several COG pathways were involved in the synthesis of fatty acids from the fermentation of protein.

**Metagenome-assembled genomes shed light on previously uncharacterized bacterial genomic diversity in the hyena gut.** Finally, to further characterize the bacterial genomic diversity present in the hyena gut, we reconstructed a total of 149 high-quality, low-contamination MAGs from the 32 metagenomes of 4 individuals (Fig. 4A; Data Set S3, Sheet 1). The average MAG completeness was 91.76% ($\pm$5.37%) and the average contamination was 1.60% ($\pm$1.24%) (Data Set S3, Sheet 1). Sizes of the MAGs ranged from 560 kb to 3.93 Mb, with a median of 1.97 Mb. The MAGs spanned 9 bacterial phyla, 12 classes, 26 orders, 50 families, and 60 genera.

All high-quality MAGs were classified to phylum, class, and order taxonomic levels (Fig. 4B). Seventy-three percent of MAGs were assigned to *Firmicutes*, 9.39% to *Bacteroidota* (formerly *Bacteroidetes*), and 8.72% to *Actinobacteriota* (formerly *Actinobacteria*). The most represented bacterial classes were *Clostridia* (45.6% of MAGs), *Bacilli* (27.5%), *Bacteroidia* (9.39%), and *Coriobacteriia* (4.69%) (Data Set S3, Sheet 1). Dominant families represented among the MAGs included *Lachnospiraceae*, *Erysipelotrichaceae*, *Anaerovoracaceae*, *Clostridiaceae*, and *Oscillospiraceae* (Fig. 4C). About 67% of our high-quality MAGs were assigned a genus, and a total of 60 genera were identified (Fig. 4B). Five of the genera represented in the MAGs came from genera that are part of the core gut microbiome in wild spotted hyenas (this study). These genera were *Bacteroides*, *Clostridium*, *Faecalitalea*, *Fusobacterium*, and *Peptostreptococcus*. In addition, 12 of the MAGs came from genera that are very abundant in the gut microbiomes of captive hyenas from zoos in China (39), specifically, the genera *Bacteroides*, *Blautia*, *Collinsella*, *Fusobacterium*, and *Peptostreptococcus*. Finally, only 20% of MAGs (31/149) were classified to species level (Fig. 4B; Data Set S3, Sheet 1). Among the MAGs classified to species were ones assigned to *Ruminococcus gnavus* (#315), *Ligilactobacillus ruminis* (#19), *Enterococcus faecalis* (#306), *Enterococcus casseliflavus* (#323), *Rhodococcus gordoniae* (#309), *Planomicrobium glaciei* (#9), and *Bifidobacterium longum* (#45) (Fig. 4A). Although some of the MAGs could be assigned to putative species, 80% of them could not because they were evolutionarily distant from the genomes in the GTDB database (Fig. S2). This suggested that there is a significant amount of genomic novelty in these MAGs (and in the hyena gut microbiome) that has not been characterized anywhere.

Of the high-quality MAGs in our data set, the one with the highest relative abundance across samples was classified as *Prevotellamassilia* sp. (#386) (Data Set S3, Sheet 2), a taxon that is closely related to an *Alloprevotella* sp. isolated from jackal guts (*Canis mesomelas*) (40) and a *Prevotella* sp. isolated from the canine gastrointestinal (GI) tract (41) (Fig. S2). A lactating, 8-year-old hyena from matriline 3 (G3) harbored the highest relative abundance of this MAG, with over 10% of their metagenomic reads mapping to this particular MAG. This same hyena a year later harbored the third-highest relative abundance of this MAG (7.6%). This hyena's grandmother, pregnant at the time, harbored the second-highest relative abundance of this MAG (6.4%). The abundance of this MAG in all other samples was <2% (Data Set S3, Sheet 2).

Several MAGs were assigned to families (e.g., *Clostridiaceae* and *Peptostreptococcaceae*) that have been reported to be more abundant in carnivores such as lions (*Panthera leo*), cheetahs (*Acinonyx jubatus*), and African wild dogs (*Lycaon picus*) compared to ruminants or other herbivores (42). In addition, several MAGs came from families (e.g., *Clostridiaceae*, *Erysipelotrichaceae*, and *Bacteroidaceae*) reported to be highly correlated with protein and fat digestibility in domestic dogs (*Canis familiarus domesticus*) (43). One MAG (#142), taxonomically assigned as *Peptostreptococcus russellii*, came from a genus of bacteria that formed part of the core microbiome in our surveyed hyenas. This MAG was closely related to a *P. russellii* isolated from a swine manure storage pit (44) (Fig. S2). Interestingly,

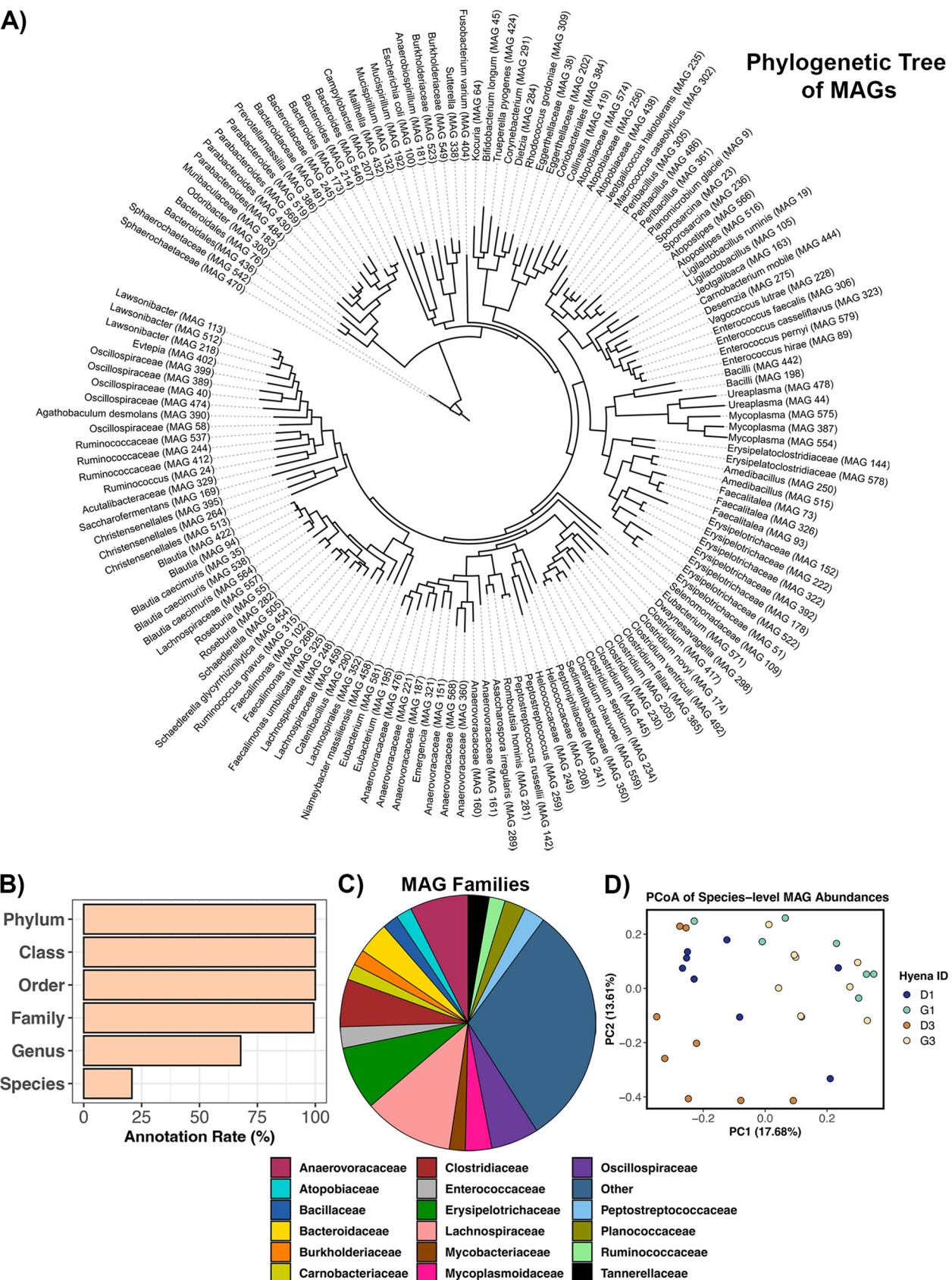

**FIG 4** A total of 149 high-quality MAGs were recovered from the hyena gut, expanding the genomic diversity currently known for carnivores. Assembled contigs were binned into MAGs with MetaBat2 and assigned taxonomy with GTDB release 202. Only high-quality MAGs (>80%

*P. russellii* sequenced the from human GI tract appears to suppress inflammation in mice (*Mus musculus*) (45).

There was one high-quality MAG, #306 (taxonomically assigned as *Enterococcus faecalis*), that was rare in the data set (0 to 0.3% relative abundance in 96% of samples) but constituted 10% of metagenomic reads in a single sample from a low-ranking, pregnant, 7-yr-old hyena (Data Set S3, Sheet 2). The fecal sample was taken during a time when prey was scarce in the Masai Mara National Reserve. The genome most closely related to this MAG was that of a pathogenic *Streptococcus* isolated from humans (Fig. S2) (46). Perhaps this hyena was suffering from a GI infection at the time of sampling. This same hyena also had the highest relative abundance of an *Enterococcus hirae* MAG (# 89) and a *Vagococcus lutrae* MAG (#228), both of which were rare in all other samples. MAGs assigned to *E. hirae* bacterial species were also recovered from the GI microbiome of healthy dogs (41).

Other MAGs to which we would like to call attention included two that were assigned to the *Spirochetes* phylum (#470 and #542). *Spirochetes* have attracted attention recently as they are found in the gut of a wide diversity of mammalian species but are absent or in low abundance in urbanized human populations (47). Another two interesting MAGs are #132 and #192, which were both assigned to the genus *Mucispirillum* from the *Deferribacteraceae* group, a group that is generally known to be capable of reducing iron, manganese, and nitrate during anaerobic respiration (48). The MAGs were found at low relative abundances in all samples, but the *Mucispirillum* genus has been previously documented in the gut microbiomes of swine, rodents, birds, and canids (49–52), although its specific relevance to host functioning needs to be investigated.

Finally, we also determined whether the relative abundances of the high-quality MAGs were associated with host factors. MAG relative abundances were significantly correlated with host identity (PERMANOVA $R^2 = 0.12$, $P = 0.002$) but were not correlated with host matriline, mean monthly prey availability, or calendar year (PERMANOVA matriline $R^2 = 0.029$, $P = 0.38$; prey $R^2 = 0.038$, $P = 0.21$; year $R^2 = 0.04$, $P = 0.08$). If we only examined the relative abundances of MAGs classified at the species level, host identity accounted for 14.68% of the variation (PERMANOVA $R^2 = 0.147$, $P = 0.001$), while the remaining predictors had no explanatory power (PERMANOVA matriline $R^2 = 0.03$, $P = 0.38$; prey $R^2 = 0.026$, $P = 0.56$; calendar year $R^2 = 0.026$, $P = 0.6$). In this ordination plot, samples appeared to cluster by hyena identity along PC axis 1 (Fig. 4D). Identical results (unpublished) were also obtained when correlating MAG abundances with the abundances of metagenome sequences that had matches to 10 common hyena prey genomes, which were calculated with sourmash (38).

## DISCUSSION

**Large-scale changes in hyena habitat are reflected in the gut microbiome.** Our data showed that the gut microbiome of 12 wild spotted hyenas was highly variable across the 23-year study period. The most extreme shift was observed in 2008 to 2009, when the nine hyenas that had samples before and after 2008 to 2009 showed marked changes in their gut microbiome compositions. This changed composition was maintained for the remainder of the study period. We observed 5-fold increases in the relative abundances of *Bacteroidales* (*Bacteroides*) and *Fusobacteriales* (*Fusobacterium*) and 10-fold decreases in the relative abundances of *Bacillales* (*Clostridiaceae* and *Peptoclostridium*), compared to earlier years. This 2-year period happened to encompass one of the most severe droughts in recent Kenyan history. Other ecosystem-wide ecological changes also occurred to the hyenas' habitat during this time. Specifically, over the 23 years of sampling, the MMNR experienced significant declines in its herbivore and lion densities and increases in livestock grazing and anthropogenic activity related to tourism (19, 27, 28). At the same time, given that hyenas are adept at adjusting to change, hyena group densities doubled. The size of the

**FIG 4** Legend (Continued)

completeness, <5% contamination) were retained. (A) Phylogenetic tree of MAGs constructed from GTDB sequence alignments and labeled with GTDB taxonomy. (B) Percent MAGs annotated at different taxonomic levels. (C) Relative abundances of MAG families. (D) PCoA results, showing the clustering of samples based on the abundances of MAGs that were classified to species level (31 MAGs).

specific social group studied here went from 60 individuals in 1990 to 120 individuals in 2013 (19). Interestingly, the sharpest increases in livestock grazing and tourism occurred during the severe drought in 2008 to 2009 (27), suggesting that anthropogenic and ecological disturbance might have lasting effects on the microbiomes of wild animals (53–58). These changes collectively may have altered the ecology of the region and affected the diet, physiology, and behavior of hyenas and, subsequently, their gut microbiome compositions. Thus, our findings show that the gut microbiome of individual hyenas may respond similarly to large-scale ecological changes. Nonetheless, the gut microbiomes of hyenas were also largely individualized outside of the synchronized large-scale shifts in composition.

**Gut microbiomes are individualized in wild spotted hyenas.** In line with results from a recently published study on the gut microbiomes of wild baboons (*Papio* spp.) sampled over a 14-year period in Amboseli National Park (14), our findings showed that each hyena exhibited largely individualized gut microbiome compositions over its adult life span. In our sampling of 12 adult female hyenas over a 23-year period, gut microbiome alpha-diversity varied with host identity. This same factor accounted for 11% of the variance in gut microbiome beta-diversity, more than was accounted for by host age, matriline, year, or prey abundance. Similarly, the gut microbiome functional profiles, i.e., the relative abundances of predicted genes and functional pathways, varied among the four hyenas that were sampled and so were the relative abundances of 149 MAGs. These findings suggested that individual hyenas likely possess unique microbiome signatures. Although our sampling was limited to 12 individuals, in many mammals, including rodents, canids, cervids, and primates, host identity is one of the primary predictors of the composition of the gut microbiome (9, 32, 59, 60). In spotted hyenas and other mammals, individualized microbiomes could arise from individual differences in social rank, immune function, reproductive state, social interactions, diet quality, stress responses, injury, and genetics (23, 61, 62). All of these variables may act individually or in concert to structure mammalian microbiomes.

We found that gut microbiome alpha-diversity moderately declined with adult age and that host age predicted 4.3% of the variance in gut microbiome beta-diversity. This was consistent with prior studies conducted in chimpanzees (*Pan troglodytes*) (63) and spotted seals (*Phoca largha*) (64), which reported a general decline in gut microbiome diversity with host aging, but our findings were not consistent with studies conducted in humans (65), cheetahs (*Acinonyx jubatus*) (66), or yaks (*Bos grunniens*) (67). In spotted hyenas from the Serengeti (68), juveniles have slightly less diverse gut microbiomes than adults, which might also be true of the population we studied, but we did not sample juveniles. Individuals undergo significant alterations to their lifestyle, diet, immune system, and overall physiology with aging, which likely impact the gut and its microbiome diversity (69, 70). Older individuals may have reduced mobility, weakened immune strength, altered gut morphology, reduced metabolic health, or infections (71, 72). Nonetheless, it is unclear what specifically contributes to the reduction in microbiome diversity as a hyena ages. This necessitates further study, given that in mammals, maintaining diverse gut microbiomes may be beneficial (73).

It was surprising that host matriline was not significantly associated with gut microbiome composition, alpha-diversity, beta-diversity, or metabolic function. Hyenas from the same matrilines did not have more similar microbiomes than hyenas from different matrilines, which was unexpected given that hyenas from the same matriline occupy similar ranks in the clan's hierarchy and are more closely genetically related than hyenas from other matrilines (20, 22). Furthermore, closely related hyenas typically spend more time together than do unrelated hyenas and occupy similar physical spaces (20, 22). Thus, we expected greater convergence in the gut microbiomes of maternally related hyenas. In this species, it appears that gut microbiome changes are more closely tied to ecological changes than to host social and genetic factors.

This plasticity and fluidity in the microbiome of wild hyenas might be a consequence of the tremendous dietary and behavioral flexibility that hyenas exhibit,

characteristics that allow them to efficiently adapt to change. First, they are opportunistic foragers with a generalist diet, can consume a wide range of species and sizes of prey, and can eat multiple if not all parts of their prey (74). They also eat insects or scavenge when ungulate prey numbers are low. Second, they live in fission-fusion societies (20, 24, 25), where the compositions of subgroups of hyenas change many times per day, allowing hyenas opportunities to interact with different members of the clan throughout the day (75). Furthermore, the dispersal of immigrant males among neighboring clans diversifies the gene pool in this population (25); immigrant males sired the majority of cubs in our studied clans and because of this, overall clan relatedness was very low (76). Hyenas also exhibit great behavioral flexibility and can adjust their behavior to improve their fitness. When anthropogenic disturbance was increasing within and outside reserve boundaries, hyenas modified their behavior to become more nocturnal and vigilant to avoid conflicts with humans (77). Perhaps the microbiome's highly variable composition in this species is a consequence of hyenas' behaviors and traits that allow them to thrive in changing environments.

**Host diet is correlated with gut microbiome variation.** Our data showed that 16S rRNA gene microbiome alpha-diversity was negatively correlated with monthly prey densities and that metagenome functional profiles were moderately correlated with this variable. It is known from earlier work (35) that the diets of MMNR hyenas change with the arrival of enormous herds of migratory wildebeest (*Connochaetes taurinus*) and zebras (*Equus quagga*) from the Serengeti; this represents a period of very high prey abundance lasting roughly 4 months each year. Hyenas of all social positions enjoy a food surplus during these months, as they have access to migratory prey in addition to the resident prey that are available year round (e.g., Thompsons's gazelle [*Eudorcas thomsonii*], impala [*Aepyceros melampus*], warthogs [*Phacochoerus africanus*], and topi [*Damaliscus lunatus*]) (34). Conversely, during periods of low prey availability, antelope prey are more scarce and more challenging to hunt, requiring greater energetic efforts from hyenas (35). Because of this, hyenas tend to scavenge more during this time and are more likely to eat desiccated carcasses and the less desirable parts of carcasses (e.g., bones, viscera). While bones are rich in fat in the marrow, they are poor in protein content. Thus, the prey species, quantity of food, and quality of diet consumed by hyenas change throughout the year, which may impact a hyena's fitness given that in this species, social rank, access to resources like food, and fitness are closely intertwined. The dietary fluctuations may potentially enrich for different microbes and metabolic pathways (e.g., bone degradation). However, we do not have any direct evidence to support this statement. These inferences go beyond the scope of our study, and future studies should examine the metabolome of wild or captive hyenas when fed high-quality versus lower-quality food.

Functional shifts in the microbiome in response to host dietary changes are not new and have been documented for other mammals. In goat kids (*Capra aegagrus hircus*), a transition from milk to a solid diet regimen is associated with linear increases in volatile fatty acid production and the upregulation of pathways involved in carbohydrate and protein metabolism (78). In captive sifakas (*Propithecus coquereli*), supplementing standard diets with diverse foliage blends versus with single plant species was associated with greater concentrations of short-chain fatty acids in the gut (79). In domestic dogs, the abundances of 15 gut metabolites differed between individuals fed a commercial diet and those fed a diet high in protein and fat and low in fiber and carbohydrates (80). At a larger scale across wild mammals, from rodents to carnivores and primates, several studies have found associations between specific microbial functions and diet guild (herbivore, omnivore, piscivore, carnivore) (81–83).

**Core, abundant, and important microbes of the hyena gut.** The core gut microbiome in hyenas was composed of 14 bacterial genera and 19 ASVs that represented 36% and 40% of the microbial community, respectively. Thus, sampled hyenas contained some of the same types of microbes, albeit at varying relative abundances. Core bacterial genera included *Alloprevotella*, *Bacteroides*, *Clostridium*, *Fusobacterium*, *Paeniclostridium*, *Peptoclostridium*, *Peptoniphilus*, and *Streptococcus*, among others. Several of these taxa include

known commensals of the mammalian GI tract, and all but three genera (*Clostridium*, *Paeniclostridium, and Streptococcus*) were among the top 10 most relatively abundant genera in the GI tract of captive spotted hyenas residing in zoos from the Shandong province in China (39). Interestingly, the gut microbiomes of the captive hyenas had elevated relative abundances of *Fusobacterium* (19.5% mean relative abundance [39], compared to 4.10% [current study]) and *Bacteroides* (8.4%, compared to 3.1% in the current study), and harbored significantly less *Clostridium* (<1% mean relative abundance versus 8.6% in the current study) compared to the wild spotted hyenas examined in the current study. This could potentially be attributed to the differences in diet and habitat between the hyenas in the two studies; the captive hyenas were all fed the same diet of rabbit meat, chicken, and beef and resided in a captive environment rather than a wild one.

More-detailed studies are required to determine whether the core genera are potentially interacting with a hyena's metabolism. We do know that *Bacteroides* forms part of the core microbiome or is highly abundant in the guts of wild cats (*Felis silvestris catus*), coyotes (*Canis latrans*), raccoon dogs (*Nyctereutes procyonoides*), black-backed jackals (*Canis mesomelas*), and myrmecophagous pangolins (*Manis javanica*) (84–87). Members of this bacterial genus are known to be functionally involved in the degradation of protein and are related to a high-fat and protein-based diet (88). *Peptoclostridium* is abundant in the gut microbiomes of captive Amur tigers (*Panthera tigris altaica*) fed a raw meat diet (89). Not much is known about *Peptoniphilus* and its role in the guts of wild mammals, but members of this genus use peptone (protein hydrolysate) as a major energy source and produce butyric acid in the process (90). We suggest that future studies examine these genera to see if they are directly or indirectly involved in helping hyenas digest their prey.

A total of 149 high-quality MAGs were recovered from the gut microbiomes of four adult female hyenas; some MAGs were assigned to genera that formed part of the core microbiome in our studied individuals. Other MAGs were classified to bacterial families or genera found in the GI tracts of other carnivores, including those of wild cats, wild dogs, and domestic dogs (42). Interestingly, 80% of our MAGs were novel, as they were not classified to the species level and were evolutionarily distant from the genomes in GTDB-r202. Similarly, a larger study found a great deal of genomic diversity in the gut microbiomes of animals from five vertebrate classes, *Mammalia, Aves, Reptilia, Amphibia*, and *Actinopterygii* (91). Those authors produced 5,596 nonredundant MAGs, of which 26% were novel and lacked a species-level match in GTDB-r89 (91). Metagenomic studies are expanding the currently known genomic diversity of the mammalian gut.

**Relatively abundant functions of the gut microbiome.** Our study did not elucidate which gut microbiome metabolic genes or pathways might be functionally important for hyenas. We showed that, not surprisingly, the most abundant bacterial functions were housekeeping functions present in virtually all bacteria. To identify the core microbial functions that characterize spotted hyenas, comparative studies are required, where the gut metagenomes of hyenas are analyzed within the context of the gut microbiome of other carnivores, herbivores, and omnivores. Only then will it become apparent which metabolic functions are up- or downregulated in spotted hyenas compared to animals consuming a different diet (e.g., herbivores), or animals with a similar diet but with distinct lifestyles and societies (e.g., less gregarious carnivores). For example, one previous study examined the functional repertoire of the gut microbiome in 77 mammalian species and found that herbivores were enriched in plant carbohydrate degradation pathways compared to carnivores and piscivores (82). In contrast, carnivore gut microbiomes were enriched in pathways related to the degradation of choline, an amine principally found in meats. We hope that follow-up studies will shed light on the uniqueness of the functional repertoire of the gut microbiome in hyenas.

**Study limitations.** We note that our study had several limitations that narrowed the overall scope of inference. First, fecal samples only came from 12 individual hyenas; thus, we could not provide a global analysis of the hyena gut microbiome, but rather that of 12 hyenas over a period of 23 years. Second, given the way the study was designed, we did not include within-matriline replicates, e.g., each matriline that was sampled only included one mother, one daughter, and one granddaughter, and only

one matriline represented each social rank category (high, medium-high, medium-low, and low). Furthermore, the number of samples from each hyena was uneven, and the time elapsed between samples was highly variable (days to years). This was due to the nature of working with wild mammals, especially wild apex carnivores like hyenas, which have large home ranges. Hyenas also exhibit fission-fusion dynamics and are often not found together at the same place and time. This makes it challenging to collect fecal samples consistently from every hyena. We might also have gone weeks without sighting the same hyenas because of our opportunistic sampling approach. Finally, metagenomic data were limited to four hyenas (32 samples total). Because of these limitations, we encourage readers to interpret our findings with caution and recognize that they apply to a small group of hyenas. In addition, it is important to note that this study made use of microbiome information in fecal samples as a surrogate for the gut microbiome. This noninvasive approach is commonly used in microbiome studies; generally, fecal samples reflect the bacteria of the colon but they are not a perfect representation of what is found in samples taken directly from the GI tract.

**Conclusions.** Using longitudinal sampling across 2 decades and multiple sequencing approaches, we found that the gut microbiomes of 12 wild female spotted hyenas were individualized and correlated with large-scale changes in the hosts' ecological environment. Gut microbiome 16S rRNA gene profiles and metagenomic functional profiles also varied with host prey density and, likely, with host diet. We also recovered 149 high-quality MAGs from the hyena gut, greatly expanding the microbial genome diversity known for hyenas and for wild mammals in general. We hope that future microbiome studies of wild, captive, or domestic mammals will employ longitudinal sampling and metagenome-based functional analyses and will reconstruct metagenome-assembled genomes from their data. Inclusion of these techniques will capture different aspects of gut microbiome variability and improve our understanding of host-microbe interactions in wild mammals.

## MATERIALS AND METHODS

**Sample and metadata collection.** The Masai Mara National Reserve (MMNR; 1,530 km$^2$) in southwestern Kenya (1°40′S, 35°50′E) is a rolling grassland habitat that constitutes the northernmost portion of the Mara-Serengeti ecosystem (92–96). The reserve has two dry seasons (late December to March and late June to mid-November) and two rainy seasons (late November to early December and April to early June) (28, 97).

Fecal samples from female members of a single social group were collected during the mornings and evenings as they were encountered. Our data set was restricted to longitudinally collected fecal samples ($n = 303$) from 12 adult (>2 years old) female spotted hyenas inhabiting the MMNR between 1993 and 2016 (Table 1; see also Tables S1 and S2 in the supplemental material). Sampling wild hyenas is difficult because of their fluid fission-fusion social dynamics and large home ranges, which make it challenging to collect samples from the same hyena at regular intervals. This combined with our opportunistic sampling meant that sample sizes were uneven for individual hyenas and the length of time in between samples was variable. Upon collection, fecal samples were stored in cryogenic vials in liquid nitrogen until being transported on dry ice to Michigan State University, wherein they were stored at −80°C until genomic DNA extractions were performed.

In the field, hyenas were identified as individuals by their unique spot patterns and sexed based on the dimorphic morphology of their erect phallus (98), and their birthdates were calculated to ±7 days based on their appearance as cubs when first observed (99). Each hyena was assigned a dominance rank based on its position in a matrix ordered by submissive behaviors displayed during dyadic agonistic encounters (21) (Table S2). In hyena societies, each new offspring inherits the rank immediately below that of its mother but above those of its older siblings. The four hyena lineages that were sampled in our study varied in their rank, with individuals from matriline 1 (M1) occupying the highest ranks in the clan's hierarchy and individuals from matriline 4 (M4) occupying some of the lowest ranks in the hierarchy. Individuals from matriline 2 (M2) were high-ranking hyenas but below all individuals from M1, and individuals from matriline 3 (M3) were low-ranking but not as low-ranking as hyenas from M4. About 77% of all samples were collected from nursing females, 10% of samples came from pregnant females, and the rest came from either nulliparous (e.g., had never given birth) or nonpregnant/nonlactating females (Table S2).

To assay prey abundance, three 4-km line-transects in the clan's territory were sampled biweekly, and all mammalian herbivores were counted within 100 m of each transect centerline. The number of herbivores was summed across the three transects, as detailed by Holekamp et al. (100). These values were averaged to calculate the mean number of herbivores counted during the 30 days prior to each fecal sample being collected (Table S2).

**DNA extractions.** Genomic DNA was extracted from the fecal samples using Qiagen DNeasy PowerSoil kits (Qiagen, Valencia, CA), following the manufacturer's recommended protocol. The order of extractions

was randomized by assigning each sample a random number without replacement and then conducting DNA extractions based on this order. Blank extraction kit controls (e.g., sterile swabs) were included to account for any background DNA contamination. The ability to PCR amplify 16S rRNA genes from samples was tested using bacteria-specific primers (8F, 5'-AGAGTTTGATCCTGGCTCAG-3'; 1492R, 5'-ACGGCTACCTTG TTACGACTT-3') and gel electrophoresis. The PCR conditions were as follows: an initial denaturation step at 95°C for 3 min, followed by 30 cycles of 95°C for 45 s, 50°C for 60 s and 72°C for 90 s. A final extension occurred at 72°C for 10 min, with a final hold at 15°C. DNA concentrations were quantified using QUBIT (Invitrogen) and ranged from 1.5 to 27.8 ng/$\mu$L for our samples (mean, 6.3 ng/$\mu$L).

**Sequencing and processing of 16S rRNA gene reads.** DNA from all fecal samples ($n = 303$) was sent for multiplexed paired-end 16S rRNA gene sequencing on the Illumina MiSeq v2 platform at the Michigan State University Genomics Core. The V4 hypervariable region of the 16S rRNA gene (250 bp) was amplified with dual-indexed, Illumina-compatible primers (515f/806r). Sequencing, library preparation, and preliminary quality filtering were completed according to the methods of Caporaso et al. (101) and Kozich et al. (102). Base calling was done by Illumina real-time analysis (RTA; v1.18.54) and output of RTA was demultiplexed and converted to FastQ format with Illumina Bcl2fastq (v2.19.0).

Raw Illumina amplicon sequence reads were processed, filtered for quality, and classified into amplicon sequence variants (ASVs) using the Divisive Amplicon Denoising Algorithm (DADA2 v1.14.1) pipeline in R (v3.6.2) (103, 104). Briefly, reads were filtered for quality, allowing for two and three errors per forward and reverse read, respectively. To remove the low-quality portions of the sequences, forward reads were trimmed to 250 bp, while reverse reads were trimmed to 220 bp. After calculating error rates, ASVs were inferred using DADA2's core denoising algorithm. Forward and reverse reads were then merged to calculate ASV relative abundances. It is important to note that the DADA2 pipeline performs merging of paired-end reads after denoising to achieve greater accuracy (104). After this, chimeric sequences were removed, leaving an average of 13,411 $\pm$ 5,431 sequences per sample. The resulting ASVs were assigned a taxonomy using the SILVA rRNA gene reference database (v132) (105), and those classified as eukarya, chloroplasts, or mitochondria were removed from the data set. Not all sequences were classified to genus or species level, and in those scenarios, the last known classification (e.g., family) was used. We exported the final ASV relative abundance table, table of ASV taxonomic designations, and sample metadata into R for statistical analysis and visualizations. These files are provided in the supplemental material (Table S2, Data Set S4) and stored in the GitHub repository for this project (see "Data availability").

Prior to further analysis, we removed two samples from the data set that had <100 sequences after processing, which left 301 samples for subsequent analyses. We used the R decontam package (v1.6.0) (106) to identify and remove contaminant ASVs based on their prevalence in control samples (DNA extracted from sterile swabs) compared to biological samples. A total of four bacterial ASVs (ASV276 *Micrococcaceae*, ASV1412 *Planococcaceae*, ASV1797 *Delftia*, and ASV1979 *Stenotrophomonas*) were present in at least 50% of control samples at relative abundances of >1%. The four ASVs had decontamination scores below our specified threshold (0.5) and were thus filtered from our data set. This left a total of 1,974 unique ASVs for analysis.

**Statistical analysis of 16S rRNA gene profiles: taxonomic composition and alpha- and beta-diversities.** In this study, we examined variation in the taxonomic composition, alpha-diversity, and beta-diversity of the hyena's gut microbiome. Unless otherwise stated, all statistical analyses and figures were performed in R (v3.6.2) (103). We first visualized taxonomic variation through stacked bar plots using the ggplot2 (v3.3.3) package (107). The plots showed the relative abundances of dominant bacterial phyla, orders, and genera across samples over the study period ($n = 301$; 1993 to 2016). Phyla with mean relative abundances of >1%, orders with mean relative abundances of >0.6%, and genera with mean relative abundances of >3% across samples were visualized in the plots, and all others were clumped into an "Other" category. These cutoffs were strictly used for visualization purposes so that taxa could be discernible in the plot.

For microbiome alpha-diversity analyses, samples were first subsampled to 2,900 reads per sample using mothur (v1.42.3) (108) to control for uneven sequencing depths. This represented the second lowest number of sequences found in our samples, and we chose this cutoff to retain as many of our samples as possible. Two samples did not meet this read number cutoff and were excluded from alpha-diversity analyses ($n = 301$ reduced to $n = 299$). Rarefaction curves of ASV richness approached saturation but were not fully saturated; thus, we estimated microbiome alpha-diversity using the Chao1 richness, Shannon diversity, and Faith's phylogenetic diversity (PD) (109–111) indices, which respectively captured microbiome richness, evenness, and phylogenetic taxonomic representation. The phyloseq package (v1.30.0) (112) was used to calculate the values for the first two metrics. The picante package (v1.8.2) (113) was used to estimate the latter metric, after supplying a phylogenetic tree of bacterial ASVs that was constructed with DECIPHER (v2.14.0) (114) and phangorn (v2.5.5) (115).

To evaluate whether host individual identity, age, matriline, average monthly prey abundance, and calendar year predicted gut microbiome alpha-diversity (Chao 1, Shannon, or PD indices on the log scale), we ran generalized linear models using the glm function from the stats package (v3.6.2) (103). After assessing model fit from residuals, we tested for statistical significance ($\alpha = 0.05$) by conducting likelihood ratio tests on all linear models using the car package (v3.0-10) (116). A second set of linear mixed models specified hyena identity and sample year as random factors and evaluated the influences of the remaining variables on gut microbiome alpha-diversity. The linear models were made with the lme4 package (v1.1-26) (117), and statistical significance was calculated as described above. Significant associations between a host factor and gut microbiome alpha-diversity were visualized via scatterplots and boxplots in ggplot2 (v3.3.3).

Microbiome beta-diversity was quantified using Jaccard distances calculated from bacterial ASV presence/absence data and Bray-Curtis distances calculated from bacterial ASV relative abundance data,

after excluding ASVs with ≤2 total reads in the data set. We also estimated weighted Unifrac distances, which considered the phylogenetic relationships among bacterial ASVs. Jaccard and Bray-Curtis distances were estimated using the vegan package (v2.5.7) (118), while Unifrac distances were generated using phyloseq (v1.30.0). To determine whether gut microbiome beta-diversity was predicted by five host factors (individual identity, age, matriline, average monthly prey abundance, and year), we performed permutational multivariate analyses of variance (PERMANOVA) tests with vegan ($n = 299$). The tests specified one type of distance matrix as the dependent variable, the five host predictors as the independent variables, and 999 permutations. PCoA ordinations based on Bray-Curtis distances were constructed in ggplot2 and were color-coded for each of the host predictors.

**Statistical analysis of 16S rRNA gene profiles: core gut microbiome.** We used the 16S rRNA gene data to identify taxa that constituted the core gut microbiome in wild spotted hyenas. For this, we identified the bacterial genera and ASVs that were present in >85% of samples at mean relative abundances of at least 0.5%. This prevalence cutoff was an intermediate of the cutoffs that have been previously employed in other mammalian microbiome studies (80% prevalence cutoff for harbor seals and 9 species of nonhuman primates, 90% prevalence cutoff for wild baboons, 100% prevalence cutoff for domestic cats and Welsh ponies) (119–123). For ASVs for which genus was unknown, the next most refined level of known taxonomic classification was used (e.g., family). Heatmaps made in ggplot2 showcased the relative abundances of core bacterial genera or ASVs for each sample. A bar graph illustrated the proportion of the gut microbiome community that was represented by the core bacterial taxa in each sample.

To determine whether the relative abundances of core bacterial genera or ASVs varied with host age, matriline, average monthly prey abundance, or calendar year, we constructed linear mixed models with the lme4 package. Host identity was set as a random effect, and only bacterial taxa with mean relative abundances of at least 1% were tested. Statistical significance of each model term was assessed by calculating P values using the Satterthwaite approximation with lmer test (v.3.1-3) (124) and applying a false-discovery rate correction. The beta-coefficients of statistically significant terms were plotted in ggplot2.

**Sequencing and processing of metagenomic reads.** To gain insight into the genomic diversity and functional potential of the gut microbiome in wild hyenas, we submitted a subset of fecal samples ($n = 32$) from two mother-daughter pairs (eight samples/hyena) for paired-end shotgun metagenomic sequencing (Table S2). These four hyenas belonged to matrilines 1 (high rank) and 3 (medium-low rank), and their samples spanned two 2-year periods: 2000 to 2001 for samples from the mothers, and 2013 to 2015 for samples from the daughters. We selected these four individuals because they had at least eight fecal samples that were collected within 2 years, were mother-daughter pairs, and represented distinct social ranks. The specific 2-year time periods selected were based on when we had overlapping samples for the two mothers and overlapping samples for the two daughters to minimize potentially confounding temporal variation.

The samples were sequenced on the Illumina HiSeq 4000 platform at the Michigan State University Genomics Core (150 plus 150 bp). Libraries were prepared using the Rubicon ThruPLEX DNA-Seq library preparation kit following the manufacturer's recommendations. Base calling was done with Illumina real-time analysis (RTA; v2.7.7), and output of RTA was demultiplexed and converted to FastQ format with Illumina Bcl2fastq (v2.19.1).

On average, samples yielded ~20 million paired-end reads (range, 14 to 23 million) with high-quality phred scores (28–30). Trimmomatic (v0.38) (125) was used to remove sequence adapters and low-quality bases from raw reads using the program's default parameters. After this filtering, samples had an average of 16,374,385 sequences ($\pm$3,049,668). Host DNA was removed by mapping sample reads to the hyena genome (126) using the graph-based aligner HISAT2 (127). Next, the forward and reverse reads for each sample were interleaved using the interleave-fastq script from the Ray assembler (v2.3.1) (128). Kraken2 (v2.1.0) was used to assign taxonomic labels to interleaved reads for each sample (129).

Interleaved reads from all samples were concatenated into a single file and assembled into contigs using Megahit (v1.2.9) (130) with default parameters. A total of 2,742,876 contigs were generated, and the quality of the assembly was evaluated using the Quality Assessment Tool for Genome Assemblies (QUAST) (v5.0.0) (131) (Table S4). To functionally annotate metagenomes, contigs were imported into Anvi'o (v.6.2) (132). Anvi'o predicted a total of 7,775,878 gene open reading frames (ORFs) using Prodigal (133). The program assigned functional annotation to genes by using the Clusters of Orthologous Groups (COGs) (134) and Kyoto Encyclopedia of Genes and Genomes (KEGG) (135) databases. To obtain an estimate of the relative abundance of each gene in a sample, quality-filtered sequences from each sample were mapped to ORFs using Salmon (v1.8.0) (136). Salmon calculated the relative abundances of ORFs in units of transcripts per million (TPM), which normalized for gene length and sample sequencing depth. Tables of the relative abundances of COG pathways and KEGG proteins are provided in the supplemental material (Data Set S2).

Contigs with a minimum length of 1,000 bp were binned into MAGs using MetaBat2 (v2.15) (137). Of the MAGs generated, 149 high-quality MAGs were obtained with completeness scores of >80% and contamination scores of <5%, as assessed by CheckM (v1.1.3) (138) (Data Set S3, Sheet 1). MAGs were assigned a taxonomy using the Genome Taxonomy Database Toolkit (GTDB-Tk) (v1.3.0) (139) with the GTDB taxonomy release 202 (Data Set S3, Sheet 1). A phylogenetic tree of MAGs was built using the multiple-sequence alignment file generated by GTDB-Tk and used the taxonomic assignments as input to RAxML (140) for refining the phylogeny. The final tree was visualized using the interactive Tree of Life (iTOL v6) (141). Individual trees of each MAG were also constructed in R to visualize the evolutionary distances between each MAG and genomes in GTDB-r202. The R package treeio (142) subsetted the large phylogeny outputted by GTDB-Tk, and ggtree was used to visualize the tree (142).

Finally, the relative abundance of each MAG in a sample was estimated using CoverM (v0.6.1) (https://github.com/wwood/CoverM) by mapping quality-filtered reads to each MAG. For every sample, CoverM output the percentage relative abundance of each MAG as well as the percentage of unmapped reads (Data Set S3, Sheet 2). On average, ~13.20% $\pm$ 7.68% of reads in any given sample (after filtering out host DNA) mapped to the MAGs (range, 1.19% to 27.42% of reads).

**Statistical analysis of metagenomic data.** Gut metagenome taxonomic profiles for the four hyenas were visualized via stacked bar plots using ggplot2, which showed the relative abundance of metagenomic reads assigned to each bacterial phylum, order, and genus as determined by Kraken2.

The predicted genes that were annotated in Anvi'o coded for 25 broad COG categories, 67 more specific COG pathways, and 7,313 unique KEGG proteins. The relative abundances (in TPM) of these functions are provided in the supplemental material (Data Set S2). TPM abundances were converted to proportions (e.g., relative abundances). We generated Bray-Curtis distances from the relative abundances of COG and KEGG functions and ran PERMANOVA tests to examine whether these were associated with four host predictors: individual identity, matriline, calendar year, or average monthly prey abundance. A hyena's age was not included as a term in the model, since the samples only spanned a total of two 2-year periods. We followed the methods described above in "Statistical analysis of 16S rRNA gene profiles: alpha- and beta-diversity." The functional categories with the highest relative abundances across samples were visualized via stacked bar plots in ggplot2.

Finally, we conducted PERMANOVA statistics on the relative abundances of 149 high-quality MAGs to determine whether these were associated with host individual identity, matriline, or mean monthly prey abundance. Tests were based on Bray-Curtis distances and used 999 permutations. The clustering of samples based on their MAG relative abundances was visualized as a PCoA ordination.

**Ethics statement.** Our research procedures were approved by the MSU IACUC on 8 January 2020 (approval number PROTO201900126) and complied with the ethical standards set by Michigan State University, the American Society of Mammalogists (69), the Kenya Wildlife Service, the Kenyan National Commission on Science, Technology and Innovation, and the Mara Conservancy.

**Data availability.** Raw sequence files were deposited in NCBI's Sequence Read Archive, under BioProject PRJNA733503 and accession numbers SAMN19468262 to SAMN19468578 (for 16S rRNA gene amplicon reads) and BioProject PRJNA734005 and accession numbers SAMN19814909 to SAMN19814940 (for shotgun metagenomic reads). All data files and R scripts for the statistical analyses and figures included in the manuscript are available on the GitHub repository for this project (https://github.com/rojascon/HyenaGutMicrobiome _AcrossGenerations).

## SUPPLEMENTAL MATERIAL

Supplemental material is available online only.

**DATA SET S1**, XLSX file, 0.2 MB.
**DATA SET S2**, XLSX file, 2.1 MB.
**DATA SET S3**, XLSX file, 0.1 MB.
**DATA SET S4**, XLSX file, 1.8 MB.
**FIG S1**, PDF file, 0.9 MB.
**FIG S2**, PDF file, 0.7 MB.
**TABLE S1**, XLSX file, 0.01 MB.
**TABLE S2**, XLSX file, 0.04 MB.
**TABLE S3**, XLSX file, 0.1 MB.
**TABLE S4**, XLSX file, 0.1 MB.

## ACKNOWLEDGMENTS

We thank the Mara Hyena Project research assistants for collecting behavioral data and fecal samples in the field and completing data entry and management in the laboratory. We furthermore thank Christina Koehler for extracting genomic DNA from fecal samples and Andrew Winters for providing guidance on lab work. We are indebted to the Kenya Wildlife Service, the Kenyan National Commission on Science, Technology and Innovation, the Kenyan National Environmental Management Authority, the Narok County Government, the Naboisho Conservancy, the Mara Conservancy, the senior warden of the Masai Mara, Brian Heath, and Christine Koshal for allowing us to conduct this research in our field site in the Masai Mara National Reserve. We are also grateful to the UNAM Institute of Ecology labs of Valeria Souza and Luis Eguiarte for hosting C.A.R. during a semester and guiding her throughout the bioinformatics processing and analysis of her metagenomes. We thank Vanja Klepac-Ceraj for providing valuable feedback on the manuscript. Finally, we thank the BEACON Center for the Study of Evolution in Action, the MSU Institute for Cyber Enabled Research, and their High Power Computer Center (HPCC) for providing technical support and the computational resources needed to complete this project.

We declare that there are no conflicts of interest.

This research was funded by the National Science Foundation (NSF) grants OISE155640, OISE1853934, DEB1353110, IOS 1755089, and OIA0939454 to K.E.H. and colleagues, the latter administered by the BEACON Center for the Study of Evolution in Action. The corresponding author, C.A.R., was also supported by a Graduate Research Fellowship from NSF, a Predoctoral Fellowship from the Ford Foundation, and a summer fellowship awarded by the Ecology, Evolution, and Behavior program at MSU.

K.R.T., K.E.H., and C.A.R. designed the study; Mara Hyena Project research technicians collected the samples and behavioral data. Our international collaborators, V.S. and M.V.J., along with J.A.E., assisted with the sequence processing and analysis of shotgun metagenomic data. C.A.R., K.R.T., and J.A.E. analyzed and interpreted all data. C.A.R. wrote the manuscript. All authors approved its final version.

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
