## [Reviewer comments · mSystems]

Taxonomic, genomic, and functional variation in the gut microbiomes of wild spotted hyenas across two decades of study

Connie Rojas, Kay Holekamp, Mariette Viladomat Jasso, Valeria Souza-Saldivar, Jonathan Eisen, and Kevin Theis

Corresponding Author(s): Connie Rojas, Michigan State University

Review Timeline:

Submission Date:

October 4, 2022

Accepted:

November 14, 2022

Editor: Sarah Hird

Reviewer(s): The reviewers have opted to remain anonymous.

Transaction Report:

DOI: <https://doi.org/10.1128/msystems.00965-22>

November 14, 2022

Dr. Connie A Rojas
Michigan State University
Integrative Biology
288 Farm Lane
East Lansing, MI 48823

Re: mSystems00965-22 (Taxonomic, genomic, and functional variation in the gut microbiomes of wild spotted hyenas across two decades of study)

Dear Dr. Connie A Rojas:

Thank you for your resubmission. The reviewers and I agree that the manuscript is an excellent contribution to mSystems.

Your manuscript has been accepted, and I am forwarding it to the ASM Journals Department for publication. For your reference, ASM Journals' address is given below. Before it can be scheduled for publication, your manuscript will be checked by the mSystems production staff to make sure that all elements meet the technical requirements for publication. They will contact you if anything needs to be revised before copyediting and production can begin. Otherwise, you will be notified when your proofs are ready to be viewed.

Publication Fees:

If you would like to submit a potential Featured Image, please email a file and a short legend to msystems@asmusa.org. Please note that we can only consider images that (i) the authors created or own and (ii) have not been previously published. By submitting, you agree that the image can be used under the same terms as the published article. File requirements: square dimensions (4" x 4"), 300 dpi resolution, RGB colorspace, TIF file format.

We recognize that the video files can become quite large, and so to avoid quality loss ASM suggests sending the video file via <https://www.wetransfer.com/>. When you have a final version of the video and the still ready to share, please send it to mSystems staff at msystems@asmusa.org.

Sincerely,

Sarah Hird
Editor, mSystems

Journals Department
Dataset S4: Accept

Table S4: Accept

Dataset S2: Accept

Table S1: Accept

Dataset S1: Accept

Table S2: Accept

Dataset S3: Accept

Table S3: Accept

Figure S2: Accept

Figure S1: Accept